# A Multi-Center, Real-Life Experience on Liquid Biopsy Practice for EGFR Testing in Non-Small Cell Lung Cancer (NSCLC) Patients

**DOI:** 10.3390/diagnostics10100765

**Published:** 2020-09-28

**Authors:** Francesco Cortiula, Giulia Pasello, Alessandro Follador, Giorgia Nardo, Valentina Polo, Elisa Scquizzato, Alessandro Del Conte, Marta Miorin, Petros Giovanis, Alessandra D’Urso, Salvator Girlando, Giulio Settanni, Vincenzo Picece, Antonello Veccia, Carla Corvaja, Stefano Indraccolo, Giovanna De Maglio

**Affiliations:** 1Dipartimento di Oncologia, Azienda Sanitaria Universitaria Friuli Centrale, 33100 Udine, Italy; alessandro.follador@asufc.sanita.fvg.it (A.F.); carlacorvaja@gmail.com (C.C.); 2Dipartimento di Medicina (DAME), Università degli Studi di Udine, 33100 Udine, Italy; 3Oncologia Medica 2, Istituto Oncologico Veneto IOV IRCCS, 35128 Padova, Italy; giulia.pasello@iov.veneto.it; 4U.O.C. Immunologia e Diagnostica Molecolare Oncologica, Istituto Oncologico Veneto IOV IRCCS, 35128 Padova, Italy; giorgia.nardo@iov.veneto.it; 5Dipartimento di Oncologia, AULSS 2 Marca Trevigiana, Ospedale Ca’ Foncello, 31100 Treviso, Italy; valentina.polo@aulss2.veneto.it; 6Dipartimento interaziendale di Anatomia Patologica, ULSS 2 Marca Trevigiana, 31100 Treviso, Italy; elisa.scquizzato@aulss2.veneto.it; 7S.O.C. Oncologia Medica e dei Tumori Immunocorrelati, Centro di Riferimento Oncologico (CRO) IRCCS, 33081 Aviano, Italy; alessandro.delconte@cro.it; 8SSD Genetica medica, Azienda Sanitaria Friuli Occidentale, Presidio Ospedaliero di Pordenone, 33170 Pordenone, Italy; marta.miorin@asfo.sanita.fvg.it; 9U.O.C. Oncologia, ULSS1 Dolomiti, Presidio Ospedaliero di Feltre, 32032 Feltre, Italy; petros.giovanis@aulss1.veneto.it; 10U.O.C. Anatomia Patologica, ULSS1 Dolomiti, Presidio Ospedaliero di Feltre, 32032 Feltre, Italy; alessandra.durso@aulss1.veneto.it; 11U.O. Anatomia Patologica, Ospedale Santa Chiara, 38122 Trento, Italy; salvatore.girlando@apss.tn.it; 12Servizio di Anatomia-Istologia Patologica, IRCCS Ospedale Sacro Cuore Don Calabria, 37024 Negrar, Italy; giulio.settanni@sacrocuore.it; 13Dipartimento di Oncologia Medica, IRCCS Ospedale Sacro Cuore Don Calabria, 37024 Negrar, Italy; vincenzo.picece@sacrocuore.it; 14U.O. Oncologia Medica, Ospedale Santa Chiara, 38122 Trento, Italy; antonello.veccia@apss.tn.it; 15SOC Anatomia Patologica, Azienda Sanitaria UniversitariaFriuli Centrale, 33100 Udine, Italy; giovanna.demaglio@asufc.sanita.fvg.it

**Keywords:** liquid biopsy, EGFR testing practice, T790M

## Abstract

Background: circulating tumor DNA (ctDNA) is a source of tumor genetic material for EGFR testing in NSCLC. Real-word data about liquid biopsy (LB) clinical practice are lacking. The aim of the study was to describe the LB practice for EGFR detection in North Eastern Italy. Methods: we conducted a multi-regional survey on ctDNA testing practices in lung cancer patients. Results: Median time from blood collection to plasma separation was 50 min (20–120 min), median time from plasma extraction to ctDNA analysis was 24 h (30 min–5 days) and median turnaround time was 24 h (6 h–5 days). Four hundred and seventy five patients and 654 samples were tested. One hundred and ninety-two patients were tested at diagnosis, with 16% EGFR mutation rate. Among the 283 patients tested at disease progression, 35% were T790M+. Main differences in LB results between 2017 and 2018 were the number of LBs performed for each patient at disease progression (2.88 vs. 1.2, respectively) and the percentage of T790M+ patients (61% vs. 26%).

## 1. Introduction

Non-small cell lung cancer (NSCLC) represents about 85% of lung cancers and adenocarcinoma is the predominant histotype [1]. Prevalence of epidermal growth factor receptor (EGFR) mutations in lung adenocarcinoma is about 15% in Caucasian patients. Exon 19 deletions and exon 21 p.Leu858ArgR substitution (L858R) account for approximately 90% of EGFR mutations. Exon 20 p.Thr790Met point mutation (T790M) represents the main mechanism of resistance to first and second generation tyrosine kinase inhibitors (TKIs), but is rarely found in treatment naïve patients [2,3]. TKIs showed an advantage in terms of survival and quality of life over chemotherapy in EGFR positive NSCLC patients and represent the standard of care in this setting [4,5,6]. Osimertinib is the treatment of choice for patients who develop T790M during TKI treatment (about 50–60% of cases) and, more recently, has shown an overall survival benefit over first generation TKIs in first line setting for patients with common EGFR mutations [7,8]. Accordingly, EGFR testing in mandatory at diagnosis for every patient with lung adenocarcinoma and for light smokers with squamous cell carcinoma (SCC), as well as for patients with progressive disease (PD) to TKIs [9]. At diagnosis, the molecular profile is determined on cytological or histopathological specimens. Circulating tumor DNA (ctDNA) analysis trough liquid biopsy represents a valid alternative, particularly if the tissue specimen is not adequate, the tissue biopsy is not feasible (e.g., bone or central nervous system localization) or it would significantly delay start of treatment [10]. Liquid biopsy is recommended also after PD for detecting T790M mutation [11]. Several studies showed that tumor DNA can be detected in the bloodstream through modern techniques, such as digital droplet polymerase chain reaction (ddPCR), allele specific PCR, BEAMing and NGS [12]. Moreover, liquid biopsy offers the chance to spare an invasive procedure and ctDNA may better reflect the tumor heterogeneity and emergent resistance mechanisms [13]. However, liquid biopsy sensitivity is about 60–70%, therefore a negative result does not exclude the presence of an EGFR mutation [14,15,16,17].

Little is known about the dynamics of tumor DNA behavior and about the determinants of its release in the bloodstream, and real-word data are lacking despite ctDNA analyses being widely used in clinical practice. In addition, different analytical methods and commercial kits have been developed. Thus, the workflow for identifying the EGFR alterations may vary among different institutes. The aim of the present study was to describe the liquid biopsy practice for EGFR detection in North Eastern Italy lung cancer centers.

## 2. Materials and Methods

We distributed a survey on ctDNA molecular testing practices in lung cancer patients and about the prevalence of EGFR mutations to the main lung cancer centers in North East Italy. The survey comprised 60 questions and was distributed in January 2019 to the chiefs of the lung cancer units involved in this study. Data were collected in March 2019. The first section of the survey explored the liquid biopsy operative procedures while the second one investigated the number of liquid biopsy tests performed on NSCLC patients and their results in terms of EGFR mutations. Overall results involved a 24 month period, between 1 January 2017 and 31 December 2018. Liquid biopsies performed both at diagnosis and at time of disease progression were included. Internal review boards of each participating center approved the survey conduction. No specific patients’ informed consent was needed for the present study since no individual patient data were collected nor reported, but we described the participating institutions’ overall liquid biopsy data (frequency of EGFR mutations and number of liquid biopsy performed).

## 3. Results

Seven major lung cancer centers in North Eastern Italy participated in the survey (Udine, Padova, Verona, Trento, Treviso, Pordenone and Feltre-Belluno). The median number of new NSCLC diagnoses per year was 130 per hospital, ranging from 100 to 337. All institutions reported EGFR as the main gene tested for clinical purposes in plasma samples. Other genes, such as KRAS and BRAF, were occasionally tested upon oncologists’ request. All centers used commercially available real-time CE-IVD tests and underwent external quality control assessment. Two institutions used Cobas EGFR Mutation Test v2 (Roche Diagnostics^®^, Basel, Switzerland), whereas the remaining five centers used Easy EGFR (Diatech Pharmacogenetics^®^, Jesi (AN), Italy). Two centers had confirmation tests’ availability: NGS or droplet digital PCR in one center each. Two centers also tested EGFR on ctDNA derived from pleural liquid and liquor. In each institution plasma separation was performed by two consequent steps. A first refrigerated centrifugation (+4 °C) at soft ramp slow speed (1200–1600 g), to avoid leucocytes lysis, and a second refrigerated centrifugation of the supernatant at ≥ 3000 g for contamination removal were performed. Extraction of ctDNA from plasma was performed by using the manual kit Helix Circulating Nucleic Acid (Diatech Pharmacogenetics^®^, Jesi (AN), Italy) with the Helix Vacuum Set (Diatech Pharmacogenetics^®^, Jesi (AN), Italy) from 1–5 mL of plasma. All participants declared that blood was collected in the same hospital where the ctDNA analysis was performed; 98% of the samples were collected on ethylenediaminetetra-acetic acid (EDTA) tubes, while 2% of the samples were collected in preservative tubes (Streck^®^, La Vista, NE, USA). Of note, Streck tubes were utilized only in one center when patients were unable to perform the blood drawing in the hub hospital. In six centers, dedicated sessions for liquid biopsy were organized in the oncological ward with a specifically trained oncological nurse performing the blood drawing. In one center, liquid biopsy was performed in the general blood collection center and EGFR testing was performed in the molecular biology laboratories of the surgical pathology unit. Median time from blood collection to plasma separation was 50 min (range: 20–120 min). Median time from plasma extraction to ctDNA analysis was 24 h (range: 30 min to 5 days). Median turnaround time (TAT), defined as the time from blood collection to the final report, was 24 h, ranging from 6 h to 5 days (Figure 1). In every institution T790M was performed both at clinical and radiological PD. Of note, in some cases, two institutions also performed liquid biopsy for T790M detection when the volume of cancer lesions increased, even before per RECIST PD. All centers declared that histo/cytological re-biopsy was suggested in case of a T790M negative liquid biopsy.

Overall, from 1 January 2017 to 31 December 2018, 475 patients were tested for EGFR on ctDNA and 654 liquid biopsies were performed. Altogether, 192 patients were tested at NSCLC diagnosis, with a 16% rate of EGFR mutation, and 283 patients were tested at disease progression. At progression, 1.63 liquid biopsies were collected for each patient and T790M was found in 35% (101/283) of the patients and in 22% (101/462) of the samples. Thirty-four percent (157/462) of liquid biopsies were negative for T790M (T790M−) and EGFR activating mutation positive (Act+), whereas 44% (204/462) were T790M−/Act−.

Main differences in liquid biopsy results between 2017 and 2018 were the number of liquid biopsies performed for each patient at disease progression (2.88 vs. 1.2, respectively) and the percentage of T790M+ patients (61% vs. 26%). On the other hand, the percentage of T790M+ samples was consistent among different years (21% in 2017 and 22% in 2018), as well as for Act+/T790− samples (33% in 2017 and 35% in 2018) (Table 1).

## 4. Discussion

The aim of the present study was to describe liquid biopsy practice and its performance in terms of EGFR activating mutations and T790M detection rate, in a real world scenario. Our survey revealed substantially homogeneous habits in liquid biopsy management for NSCLC in first and second generation TKI progression setting among the interviewed centers, in accordance with national and international guidelines. Notably, every center processed the blood sample for plasma extraction within 2 h, as recommended to preserve ctDNA integrity [11,18]. Main differences were the different kits used and TAT, which varied from 24 h to 5 days. T790M positivity ranged from 29% to 86% among the centers, probably due to selection bias, considering the retrospective nature of the study. Operative differences, despite being minimal, may also have contributed. The referral of patients from spoke centers, when tissue biopsy was unfeasible, may explain the high number of liquid biopsies performed at diagnosis (29% of the total). EGFR positive diagnostic liquid biopsies were 16%, consistently with literature data on EGFR prevalence, thus confirming the reliability of liquid biopsy as a diagnostics tool [19]. Overall, our data showed T790M in 35% of patients and in 22% of samples for liquid biopsies performed after PD. Randomized clinical trials reported T790M in about 50–60% of the patients, using both liquid and tissue biopsies [7,20]. Real word data reported a T790M detection rate through liquid biopsy of about 25%, using real-time PCR based tests, while it reached 66% using ddPCR [21,22,23,24]. Performing liquid biopsy at first hints of possible TKI resistance, and performing before clinical or canonical radiological PD may have further lowered T790M detectability in our cohort [21]. In the present survey, repeating liquid biopsy in case of a first negative result emerged as a common practice (1.63 biopsies for each patient). This reflects the intent of avoiding more invasive procedures and it may depict an excess of entrusting liquid biopsy. The decrease of liquid biopsy per patient from 2017 to 2018 could be explained with the publication in 2018 of national and international guidelines, which strongly recommend performing a solid biopsy after a first negative liquid biopsy [10,18]. Notably, the number of T790M+ patients decreased from 2017 to 2018 (61% vs. 26%), along with the number of liquid biopsies performed for each patient (2.88 in 2017 and 1.2 in 2018) (Table 1). Thus, repeating a liquid biopsy might have a clinical rationale and could increase its overall sensitivity for detecting resistance mutations [25]. To date, only the presence of extra thoracic disease, especially with bone localization, has proved to be a clinical positive predictive factor for T790M detection by liquid biopsy [26,27], whereas exclusive intracranial disease correlates with a low level of ctDNA [28]. Other questions still unanswered are whether increasing sensitivity by introducing more sensitive assays would be clinically meaningful, and whether an EGFR mutation with low mutated allelic fraction would still be predictive of TKI efficacy [29]. Main limitations of our study are its retrospective nature, the lack of single patient longitudinal information and the lack of information about the number and results of tissue biopsy performed.

## 5. Conclusions

The present survey showed a substantial concordance in the liquid biopsy clinical practice among seven different lung cancer centers in Italy, in accordance with main international guidelines. Comparing real-life experiences is of paramount importance in order to optimize the clinical practices, considering that awareness and knowledge about liquid biopsy use are still limited [30], and to keep up to date with a rapidly evolving molecular testing scenario.

## Figures and Tables

**Figure 1 diagnostics-10-00765-f001:**
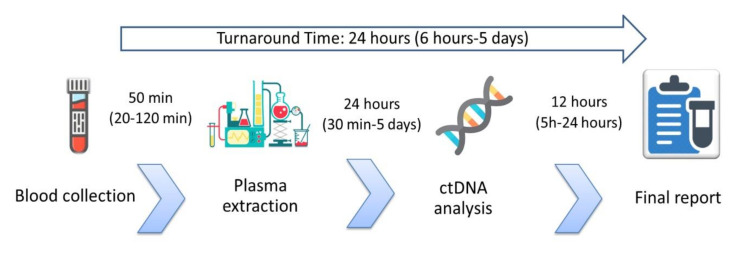
Turnaround time and workflow steps timing. Data are reported as median (range).

**Table 1 diagnostics-10-00765-t001:** Number of patients tested and liquid biopsy results.

Year	Overall	2017	2018
Pts. tested (overall)	475	159	316
LB (overall)	654	298	356
At diagnosis			
Pts tested	192	84	108
LB performed	192	84	108
% of EGFR mutation	16%	18%	15%
At PD			
Pts tested	283	75	208
LB performed	462	214	248
LB/pts ratio	1.63	2.88	1.2
% of pts T790M+	35%	61%	26%
% of LB T790M+	22%	21%	22%
% of LB Act+/T790M−	34%	33%	35%
% of LB Act−/T790M−	44%	45%	43%

Abbreviations: LB = liquid biopsy, EGFR = epidermal growth factor receptor, PD = disease progression, Pts = patients, Act + = EGFR activating mutation positive.

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
