# Peer review of "A Multi-Center, Real-Life Experience on Liquid Biopsy Practice for EGFR Testing in Non-Small Cell Lung Cancer (NSCLC) Patients"

_diagnostics, 2020, doi:10.3390/diagnostics10100765_

Round 1

Reviewer 1 Report

The authors described the real-world utility of EGFR-liquid biopsy analysis in the clinical practice, in different North Eastern Italy lung cancer centers, comparing some methodological and practical aspects. They found a general concordance from the different centers. Although results of the study not evidence particular novel data, they highlight some practical aspects of the clinical practice that could be useful for the readers.

Some points should be more specified:

1)      Could the authors specify which kit is used for plasma extraction and DNA separation and whether it differs among the centers involved in the survey?

2)      Could the authors explain why 10% of samples were tested on streck tubes instead of EDTA? 

Author Response

Point 1: Could the authors specify which kit is used for plasma extraction and DNA separation and whether it differs among the centers involved in the survey?

Response 1: Thank you for your comment.  In each Institution plasma separation was performed by two consequent steps, accordingly to national guidelines [1]. A first refrigerated centrifugation (+4°C) at soft ramp slow speed (1200-1600g), to avoid leucocytes lysis, and a second refrigerated centrifugation of the supernatant  at ≥3000g for contaminations removal were performed. CtDNA extraction from plasma  was performed by using the manual kit Helix Circulating Nucleic Acid (diatech pharmacogenetics) with the Helix Vacuum Set (diatech pharmacogenetics) from 1-5mL of plasma. We have implemented the manuscript with this information.

Point 2: Could the authors explain why 10% of samples were tested on streck tubes instead of EDTA?

Response 2: Thank you for your comment. Streck tubes were used only in one center and involved about 3% of the patients included in the present study. They were utilized when the patients were unable to perform the blood drawing in the hub hospital. We have implemented the manuscript with this information.

Kind regards

Reviewer 2 Report

While more patients received liquid biopsy in 2018 than 2017, the number of liquid biopsies performed in each patient decreased. The authors cited the guideline that recommend tissue biopsy for the reason. Along with this, the T790M detection rate by liquid biopsy decreased, suggesting that multiple liquid biopsies are associated with the T790M detection rate. However, in patients in whom the T790M gene mutation was not detected in liquid biopsies, the T790M gene mutation might be detected by tissue biopsy. As authors stated, one of the limitations of this study is the lack of information on tissue biopsy.

It is considered that there is no particular point that needs to be revised.

Author Response

Point 1: While more patients received liquid biopsy in 2018 than 2017, the number of liquid biopsies performed in each patient decreased. The authors cited the guideline that recommend tissue biopsy for the reason. Along with this, the T790M detection rate by liquid biopsy decreased, suggesting that multiple liquid biopsies are associated with the T790M detection rate. However, in patients in whom the T790M gene mutation was not detected in liquid biopsies, the T790M gene mutation might be detected by tissue biopsy. As authors stated, one of the limitations of this study is the lack of information on tissue biopsy.

It is considered that there is no particular point that needs to be revised.

Response 1: Thank for your comment. Probably in the next future we should start to consider liquid biopsy and tissue biopsy as complementary, instead of alternative, diagnostics tools for a better NSCLC management.

Kind regards